# The Non-Invasive Assessment of Circulating D-Loop and mt-ccf Levels Opens an Intriguing Spyhole into Novel Approaches for the Tricky Diagnosis of NASH

**DOI:** 10.3390/ijms24032331

**Published:** 2023-01-24

**Authors:** Erika Paolini, Miriam Longo, Alberto Corsini, Paola Dongiovanni

**Affiliations:** 1General Medicine and Metabolic Diseases, Fondazione IRCCS Cà Granda Ospedale Maggiore Policlinico, Via Francesco Sforza 35, 20122 Milan, Italy; 2Department of Pharmacological and Biomolecular Sciences, Università degli Studi di Milano, 20133 Milano, Italy; 3Department of Clinical Sciences and Community Health, Università degli Studi di Milano, 20122 Milano, Italy; 4IRCCS Multimedica, 20099 Milan, Italy

**Keywords:** NAFLD, NASH, mitochondrial dysfunction, biomarkers, D-loop, mt-ccf

## Abstract

Nonalcoholic fatty liver disease (NAFLD) is the commonest liver disease worldwide affecting both adults and children. Nowadays, no therapeutic strategies have been approved for NAFLD management, and hepatic biopsy remains the gold standard procedure for its diagnosis. NAFLD is a multifactorial disease whose pathogenesis is affected by environmental and genetic factors, and it covers a spectrum of conditions ranging from simple steatosis up to nonalcoholic steatohepatitis (NASH), fibrosis, cirrhosis, and hepatocellular carcinoma (HCC). Several studies underlined the urgent need to develop an NAFLD risk prediction model based on genetics, biochemical indicators, and metabolic disorders. The loss of mitochondrial dynamics represents a typical feature of progressive NAFLD. The imbalance of mitochondrial lifecycle together with the impairment of mitochondrial biomass and function trigger oxidative stress, which in turn damages mitochondrial DNA (mtDNA). We recently demonstrated that the main genetic predictors of NAFLD led to mitochondrial dysfunction. Moreover, emerging evidence shows that variations in the displacement loop (D-loop) region impair mtDNA replication, and they have been associated with advanced NAFLD. Finally, lower levels of mitophagy foster the overload of damaged mitochondria, resulting in the release of cell-free circulating mitochondrial DNA (mt-ccf) that exacerbates liver injury. Thus, in this review we summarized what is known about D-loop region alterations and mt-ccf content during NAFLD to propose them as novel non-invasive biomarkers.

## 1. Introduction

Over the past two decades, nonalcoholic fatty liver disease (NAFLD) has raised the burden on healthcare worldwide, affecting both children and adults. In 2016, the global NAFLD prevalence was 25%, with an increase around 30% in 2019 and the highest being reported in the Middle East (32%) and South America (31%), followed by Asia (27%), USA (24%), and Europe (23%), with the lowest in Africa (14%) [1]. NAFLD encompasses an umbrella of chronic liver disorders, ranging from simple steatosis to nonalcoholic steatohepatitis (NASH) and fibrosis. In 10–20% of cases, NASH could progress up to cirrhosis and hepatocellular carcinoma (HCC), and nowadays, it represents the third leading cause of liver transplantation (LT) [2]. NAFLD epidemiology is closely interconnected with epidemic obesity and type 2 diabetes mellitus (T2DM), and it is related to the *spectrum* of metabolic syndromes including insulin resistance (IR), dyslipidemia, hyperglycemia, hypertension, and cardiovascular disease (CVD) [3]. Sedentary lifestyle coupled with a hypercaloric diet are the major risk factors that influence visceral adiposity and the development of peripheral IR. Furthermore, NAFLD exhibits a strong heritable component and polymorphisms in the *Patatin-like Phospholipase Domain-containing 3* (*PNPLA3*), *Membrane Bound O-acyltransferase Domain-containing 7* (*MBOAT7*), and *Transmembrane 6 Superfamily Member 2* (*TM6SF2*) genes, which lead to hepatic fat accumulation and represent the most robust genetic modifiers known to affect NAFLD susceptibility [4,5,6]. The mechanisms underlying NAFLD pathogenesis are highly complex, and parallel hits contribute to the disease onset and progression. It has been largely described that mitochondria play a key role in the switching towards NASH to the point that NAFLD may be dubbed “mitochondrial disease”. The liver is enriched in mitochondria which actively contribute to energy demand and body homeostasis. In the early stages of NAFLD, in response to the exacerbated lipid overload, the liver engages in a strategy called “mitochondrial flexibility” through which these organelles modify their number, mass, and activity by modulating their lifecycle and turnover, which encompass fusion, fission, and mitophagy. Nevertheless, this adaptability is lost during NAFLD progression to NASH, resulting in the accumulation of damaged mitochondria with irregular shape, which parallels impaired oxidative phosphorylation (OXPHOS) capacity and reduced β-oxidation [7,8]. Regarding this process, Leveille and colleagues showed that both mice and humans with NASH exhibit blunted ketogenesis, maximal respiration, and Krebs cycle [9].

In both NAFLD and NASH patients, an increased mitochondrial mass together with the assembly of megamitochondria have been observed [10]. Notably, the impaired mitochondrial biomass spills over the organelles’ number and function, and it may correlate with alterations in the displacement loop (D-loop) region of the mitochondrial DNA (mtDNA), which is the master regulator of mtDNA replication and transcription [11]. Nucleotide D-loop variations have been observed in 150 biopsy-proven NASH patients compared to 150 healthy individuals. Specifically, the m.16129 AA and m.16249 CC genotypes were associated with advanced stages of fibrosis and lobular inflammation, respectively [12]. Emerging evidence underlines that alterations in mtDNA trigger tumorigenesis, thus prompting the possibility to consider the mtDNA as a potential predictor of advanced hepatic injury [13]. In addition, the sustained mitochondrial activity failure boosts oxidative stress and inflammation, along with the release of mitochondrial-damage-associated molecular pattern (mito-DAMPS) from harmed mitochondria. Among mito-DAMPS, the cell-free circulating mtDNA (mt-ccf) is the major component and may interact with toll-like receptor 9 (TLR9), thereby activating inflammation and fibrosis [14]. High-fat-diet (HFD)-fed mice and NAFLD/NASH patients consistently displayed higher serum concentrations of mito-DAMPS, especially in the presence of fibrosis. Furthermore, the higher mt-ccf levels mirrored the hepatocellular injury and were associated with NASH and fibrosis in both rodents and humans [14,15]. Increased serum mt-ccf content was found even in T2DM patients with nucleotide variations in the D-loop region, suggesting a linkage between D-loop mutations and the damaged-mitochondria-driven release of mtDNA fragments [16]. Current associations among mtDNA mutations and NAFLD stages are listed in Table 1.

To date, hepatic biopsy represents the gold standard for the diagnosis of NASH. Facing the growing prevalence of NAFLD, non-invasive methodological approaches via circulating molecules and ultrasound-based transient elastography have been proposed to estimate the severity of liver disease, although we are still far from the identification of biomarkers which enable the diagnosis of NASH [24]. As mentioned above, mitochondria are gaining an increasingly prominent role in NAFLD development, as their failure orchestrates the switch up to HCC, and mtDNA seems to contribute to NAFLD progression. Therefore, this review aims to summarize the role of D-loop and mt-ccf in the large *spectrum* of NAFLD pathophysiology, thus pinpointing them as novel diagnostic biomarkers.

## 2. NAFLD Diagnosis and Prognosis: Current Drawbacks and Newly Fashioned Strategies

The prevalence of NAFLD is increasing worldwide in parallel with obesity and T2DM. The progression of steatosis towards fibrosis or cirrhosis enhances the risk of HCC development, and nowadays, the United Network for Organ Sharing reveals that the number of NASH adults listed for LT exceeds those with chronic hepatitis C [25]. Currently, no therapeutic approaches are available for clinical management and prevention of NAFLD. Lifestyle interventions comprising a low-calorie diet and physical activity are the pillars for NAFLD treatment. It has been noted in NASH subjects that weight loss of 5–7% and 8–10% is required to improve hepatic steatosis and inflammation/fibrosis, respectively [26]. Nevertheless, the scant compliance of patients with lifestyle interventions has outlined the urgent need of NAFLD pharmacotherapy.

Hepatic biopsy is considered the gold standard procedure for the diagnosis and staging of NAFLD, which can delineate the degree of steatosis, necroinflammation activity, and stage of ballooning [27]. Nevertheless, it exhibits several limitations due to its low applicability in some patients and children, the high cost, sample size, and repeatability, which in turn could worsen the diagnostic accuracy. Thus, the European Association for the Study of the Liver, the European Association for the Study of Diabetes, and the European Association for the Study of Obesity (EASL-EASD-EASO) recommended several non-invasive techniques to diagnose NAFLD [28]. Among them, the transient elastography known as Fibroscan results in the most accurate non-invasive imaging method to identify patients at risk of advanced fibrosis by evaluating liver stiffness through the controlled attenuation parameter (CAP) [29]. Notwithstanding, the assessment of liver stiffness is not always successful in obese patients or the subset of individuals with T2DM, hypertension, and higher levels of transaminases, causing the consequent overestimation of liver fibrosis. Therefore, to overcome the use of histological evaluation, it becomes crucial to identify novel non-invasive biomarkers with prognostic value for the assessment of progressive liver damage [30,31].

Moreover, most patients are asymptomatic during NAFLD onset, and early diagnosis is useful to prevent the worsening to NASH-associated morbidity and mortality. In this regard, impending evidence highlights the role of mtDNA in mitochondrial derangements, neurodegenerative disorders, and cancer by focusing on the upregulation of mtDNA copy number alongside mitochondrial biogenesis as a compensatory mechanism to sustain cellular bioenergetics [7,8]. Notoriously, mitochondria play an ascertained role in NAFLD progression through the metabolization of glucose and lipids by OXPHOS and β-oxidation, which are blunted during disease progression, triggering oxidative stress. Consequently, reactive oxygen species (ROS) promote inflammation and fibrosis alongside mtDNA damage, concomitantly with the increased release of mt-ccf from impaired mitochondria [32,33]. In a cohort of 318 NAFLD patients, a strong association has been described between higher mtDNA copies, which disclose the number or mass of mitochondria, and oxidative stress, which leads to inflammation, thus emphasizing the correlation between mtDNA alterations and severity of liver injury [34]. Furthermore, high serum levels of mt-ccf were detected in HFD-fed mice and NASH patients, thus supporting the hypothesis that they may trigger the inflammatory response and promote the switching to cirrhosis and HCC [14]. Lastly, An et al. revealed that in three inbred mouse strains undergoing chronic administration of hepatotoxin thioacetamide, the mito-DAMPs released from injured hepatic mitochondria directly activated hepatic stellate cells, driving liver scarring. In addition, these authors found that circulating mito-DAMPs were markedly increased in 20 patients with NASH and fibrosis [15]. Therefore, it could be speculated that high mtDNA copy number, which could be evaluated by amplifying a fragment of mtDNA inside tissues (i.e., D-loop), may reflect the increased mitochondrial biomass or number, but it may not be representative of an enhanced mitochondrial function. Conversely, high levels of mt-ccf, which is released from damaged mitochondria and can be detected in serum, may better estimate the state of mitochondrial activity, supporting the assumption that both mtDNA content and mt-ccf could be promising strategies for identifying patients at risk of progressive NAFLD.

### Genetics and Metabolic Factors Co-Aid NAFLD Diagnosis

Environmental factors, including IR and obesity, strictly influence the NAFLD pathogenesis. Nevertheless, NAFLD exhibits a strong heritable component, and in recent decades, candidate gene studies and genome-wide association studies (GWASs) highlighted the impact of single genetic variants on progressive NAFLD. To date, it is preferred to combine individual loci into polygenic risk scores (PRSs) to foresee the risk of developing advanced disease [35,36]. PRS performance should be attested by using receiver operating characteristics (ROC) curves and analyzing the area under the curve (AURO), possibly combining them with metabolic parameters to obtain higher diagnostic performance in predicting NAFLD [37]. We firstly performed a randomization analysis and built a PRS to demonstrate that the impact of *PNPLA3*, *TM6SF2*, and *MBOAT7* risk alleles on hepatic injuries is directly proportional to their ability to promote hepatic fat deposition, which is the main driver of NAFLD severity [38].

Hepatic fat content variability was also explained for 8.7% by metabolic factors and for 16.1% by inherited variations in overweight children with NAFLD, and this result was confirmed in 2042 pediatric patients in whom the combination of body mass index (BMI), insulin levels, and *PNPLA3* and *TM6SF2* genetic variants in a PRS seems to be more trustworthy for predicting fatty liver compared to the method based only on BMI and insulin [39,40]. We further demonstrated in a large NAFLD cohort (*n* = 2556) that PRS predicted HCC more accurately than single variants and may help in stratifying HCC risk in individuals with dysmetabolism [41].

In 1380 biopsy-proven NAFLD patients, we recently showed that the co-presence of PNPLA3 I148M, *MBOAT7 rs641738*, and TM6SF2 E167K genetic variations was associated with advanced liver disease and HCC development independently of higher transaminases levels, cholesterol, and BMI, thus corroborating their synergistic role in progressive NAFLD [6]. Moreover, *TM6SF2* and *MBOAT7* silencing through Clustered-Regularly-Interspaced-Short-Palindromic-Repeats (CRISPR)-associated protein 9 (CRISPR/Cas9) in HepG2 cells, which are homozygous for the I148M PNPLA3 mutation, led to fat accumulation and an increased number of mitochondria, suggesting an imbalance in mitochondrial dynamics associated with impaired mitochondrial activity with a shift toward anaerobic glycolysis [6].

To conclude, robust prediction models should integrate a larger number of rare and common variants and other dynamic risk factors, among which mitochondrial D-loop and mt-ccf levels could be included, thus improving the prediction of advanced liver damage [42].

## 3. Mitochondria as the Hepatocytes’ “Powerhouse”: How do They Contribute to NAFLD?

The liver harnesses around 15% of the organism’s oxygen under physiological conditions; therefore, it is enriched in the number of mitochondria, ranging from 400–500 per hepatocyte. Mitochondria are highly dynamic organelles characterized by a double-membrane structure enclosing a dense matrix and are essential for cell homeostasis and its metabolic activity, as they provide energy through the mitochondrial OXPHOS system and ATP synthesis, as well as regulate redox status, β-oxidation, the tricarboxylic acid cycle (TCA), ketogenesis, glucose, and lipid metabolism [43]. Indeed, the mitochondrial matrix is filled with enzymes that participate in β-oxidation and TCA alongside several circular mtDNA double-strand copies, which encode 13 subunits of the OXPHOS respiratory chain, together with two ribosomal RNAs (rRNAs) and twenty-two transfer RNAs (tRNAs) required for their translation via mitochondrial protein synthesis [44]. However, the major part of the mitochondrial proteins’ origin from nuclear genes and several nuclear-encoded factors are involved in the assembly of mtDNA binding core proteins (nucleoids) that package mtDNA [45]. Human mtDNA contains even a non-coding region between the phenylalanine and proline mt-tRNAs and holds heavy-strand promoters (HSPs) and light-strand promoters (LSPs) for transcription of the heavy and light strands, respectively, as well as the origin of heavy-strand replication, O_H_. The main part of the non-coding region includes a linear third DNA strand assembling a D-loop structure, whose variations have been recently associated with mitochondrial abnormalities and related disorders [46,47].

The mitochondrial lifecycle, known as mitobiogenesis, encompasses fusion, fission, and mitophagy processes and is regulated by *Peroxisome proliferator-activated receptor gamma coactivator-1* (*PGC-1α*) alongside the replication and transcription of mtDNA and the assembly of OXPHOS complexes. The fusion process sustains the OXPHOS capacity and the elongation of mitochondria by merging mitochondrial outer membranes (MOMs), which define the intermembrane space and encircle several proteins, including the Mitofusin ½ (Mfn1 and Mfn2) and Fission 1 (Fis1), which orchestrate fusion and fission, respectively. Conversely, during fission, one mitochondrion is separated into two or more organelles by dynamin-related protein 1 (DRP1), which is recruited on MOMs by Fis1. On the other side, the inner mitochondrial membranes (IMMs) delimit the mitochondrial matrix and hold proteins comprising ATP synthase (ATPase), complexes I, II, III, and IV of the respiratory activity chain, and optic atrophy-1 (Opa1), which merges the IMMs (25). The impairment of fusion and fission mechanisms leads to the accumulation of damaged and non-functional mitochondria. The latter may be exacerbated by a hampered mechanism of mitophagy, which physiologically provides the disruption of damaged mitochondria [7,8,48]. Additionally, the loss of mitochondrial plasticity favors the onset of the *Warburg* effect, through which the hepatocytes promote the switch toward anaerobic glycolysis, even in the presence of oxygen, to sustain the energy demand [49].

Mounting evidence underlines that the loss of mitochondrial flexibility paralleling the failure of OXPHOS capacity strongly impacts NAFLD susceptibility, from the early stages to advanced liver damage [7,8,50,51,52]. The collapse of mitochondrial activity is correlated with fusion–fission imbalance, which affects the mitochondrial architecture and promotes the assembly of giant mitochondria, which have been observed in both adult and pediatric patients with NASH [10]. In this regard, we recently reported the case of a 40-year-old woman with aggressive NAFLD due to severe hypertriglyceridemia that ensued from a combination of genetic variants and metabolic risk factors. By exploiting transmission electron microscopy (TEM), we observed hepatic giant mitochondria with atypical membrane distribution, irregular cristae in terms of number and dimension, and large dark inclusions, thus concluding that lipid accumulation in hepatocytes together with mutations in genes involved in mitochondrial function may affect mitochondrial structure during NAFLD [53].

The compromised mitochondrial dynamics and activity prompt ROS production, and persistent oxidative stress has been associated with increased mtDNA copy number and with both nuclear and mitochondrial DNA damage. Indeed, in a genetic in vitro model of NAFLD, ROS accumulation was correlated with a higher frequency of apurinic/apyrimidinic (AP) sites in the nuclear genome, while a large amount of mtDNA injury has been described in patients with severe NASH compared to those without advanced liver disease [6,52,54], suggesting that acquired mutations in mtDNA may occur over the course of the disease. As such, these mutations are progressively gaining attention for NALFD assessment. Interestingly, the maternal inheritance of mitochondria is the same in the genome of all subjects, except the mtDNA mutations spilling the heteroplasmy. This leads to a heterogeneous population of mtDNA within the same cell, and even within the same mitochondrion [55,56]. In NASH, mtDNA incurs a high mutational rate and degree of heteroplasmy, which may modify the total mtDNA content. The concept of heteroplasmy in NASH is a hot topic, though still in the early stages of research, and a few studies have identified heteroplasmic mutations in the hepatic mtDNA sequence, proving the need to deepen this area of research. Sookoian et al. have identified a higher frequency of heteroplasmic regions in the cytochrome *c* oxidase, cytochrome *b*, NADH dehydrogenase, and other members of the OXPHOS complexes occurring in NASH patients with the concomitant increase in DNA oxidative adducts and lipid peroxyl radicals, compared to those with less disease activity [18,22,57]. It has even been reported that cancers may have a specific pattern of mtDNA mutations and heteroplasmy, and to date, the advances in next-generation sequencing (NGS) technologies have provided wide and in-depth coverage methods to identify the tissues-specific profile of mtDNA heteroplasmy [58]. Nonetheless, this field is still uncharted, and a limited number of studies have explored the heteroplasmy in HCC. Among them, Li and collaborators have analyzed *n* = 88 HCC samples and pinpointed 202 tumor-specific heteroplasmic mutations, many of which were localized in the NADH dehydrogenase genes (*ND1*, *ND3*, *ND4*, *ND5*, *ND6*), thus suggesting that the latter might have a pivotal role in tumorigenesis and offering new insights for further understanding of liver cancer development [21,23].

## 4. Brief Overview on Mitochondrial DNA Content as Biomarker of Mitochondrial Mass

Oxidative stress fulfils a key role in several common disorders including diabetes, CVD, neurodegenerative disease, cancer, and renal and hepatic disorders. Particularly, the imbalance of the redox status, which is the main form of intracellular signaling for energy production, impairs mitochondrial function, causing damages to proteins, lipids, and DNA, both nuclear and mitochondrial [59].

Qualitative and quantitative aberrances in mtDNA were correlated with several metabolic disorders and, albeit the association between mtDNA damage and oxidative stress in NAFLD development is consistent, the prognostic role of mtDNA copy number needs to be further elucidated. As outlined above, mtDNA contains the non-coding D-loop region, which regulates its replication and transcription, and some evidence indicates the role of D-loop content and variations in NASH [12,17]. It has been found that increased hepatic mtDNA copies led to mitochondrial bioenergetic deficit in mice and patients with fibrosis and inflammation, supporting the aforementioned concept that the augmented mitochondrial biomass does not necessarily mirror the improved mitochondrial activity [17,60]. An increased mitochondrial biomass together with raised mtDNA copy number were even described in human lung fibroblasts treated with hydrogen peroxide to trigger oxidative stress [19]. Lee and colleagues demonstrated that oxidative damage led to increased mitochondrial biogenesis and mtDNA content, and this result is consistent with that derived from a study including 150 biopsy-proven NASH patients in whom there was a correlation between oxidative stress and higher levels of mtDNA [12,61]. Likewise, a case-control study carried out by Kamfar et al. assessed mtDNA content in fresh liver biopsies of 43 NAFLD subjects and 20 bariatric patients without NAFLD. The authors showed that the amount of mtDNA was an independent predictor of NAFLD susceptibility, and its copy number was more than three times higher in both patients affected by simple steatosis or NASH compared to normal livers [17].

Changes in mtDNA content have also been described in several types of cancer, although data are still conflicting. Increased mtDNA was detected in whole blood or saliva of patients at risk of breast and neck cancer, respectively [62,63]. Conversely, a Chinese study revealed that patients with breast cancer displayed a negative correlation between tissue mtDNA content and tumor progression [64]. In keeping with this evidence, Fan et al. found that the 82% of patients with breast cancer exhibited lower levels of mtDNA compared to the adjacent tissue [65]. Concerning HCC, two different studies demonstrated a strong association between carcinogenesis and mtDNA mutations related to the D-loop region in 17/50 and 13/19 patients, respectively [66,67]. Wong et al. found that mtDNA copy number in the tumor tissues of 20 HCC patients was higher compared to those without mtDNA mutations, thus suggesting that alterations in the D-loop region affect mtDNA content by adapting the mtDNA replication and transcription and enhancing the negative correlation between the higher number of mitochondria and their functionality [20].

Finally, aberrances in mtDNA copy number were described in metabolic disorders such as T2DM and obesity, showing new contrasting evidence. Lee and colleagues proposed lower levels of mtDNA in peripheral blood as a prognostic marker for the development of T2DM alongside another study in a Chinese population [68,69]. Cho et al. assessed mtDNA copy number in peripheral leukocytes of 1108 participants at risk of T2DM. They did not find alterations of mtDNA content in four follow-ups spanning 8 years, but they showed that the integration of mtDNA content evaluation to clinical data (i.e., age, sex, BMI, hemoglobin A_1c_) and the results of oral glucose tolerance tests augmented diabetes prediction [70].

Nonetheless, higher copies of mtDNA were detected in peripheral blood cells of patients with T2DM, in diabetic subjects with nephropathy, and in NAFLD individuals [34,70,71,72]. Taken together, these controversial studies indicate the emerging role of mtDNA in the pathogenesis of several diseases, albeit its adaptation among different disorders needs to be further elucidated. To summarize, this chapter aimed to underline the correlation between mtDNA content and the severity of different metabolic diseases, highlighting that the assessment of mtDNA in tissues or biological fluids may provide a novel non-invasive methodological approach to achieve a thorough diagnosis.

## 5. The Prognostic Power of Mitochondrial D-Loop Region during NAFLD

Mitochondria provide their own double-strand genome including the machinery for replication, transcription, transcript maturation, and mitochondrial translation [73]. The replication of mtDNA is independent of the entire cell cycle and depends on the origins of replication O_h_ and O_L_, which are related to the heavy and light strand, respectively. The replication begins at O_h_ in unidirectional mode until two-thirds of the mitochondrial genome is replicated. Subsequently, the replication moves on to O_L_, which is triggered to assemble the stem–loop structure and proceeds to form a double-strand DNA [74,75]. As outlined above, the mtDNA contains a unique D-loop structure, which is an additional linear third strand that controls the mtDNA transcription and replication. The D-loop was first detected in electron micrograph images in mouse and chicken mtDNA, and it is present in the mtDNA of several animal species such as rabbits, cows, and *xenopus*, as well as humans [76]. Holt et al. firstly hypothesized that the D-loop region may regulate mitochondrial replication and transcription by exploiting the ATPase protein, which recruits the D-loop and promotes the interaction and separation of mtDNA molecules anchored to IMMs. Thus, the D-loop acts as a cis-element that orchestrates the packaging of the mtDNA genome [77].

It has been established that mtDNA is susceptible to mutations caused by oxidative stress, especially in the site downstream of the 3′ end of the non-coding hypervariable D-loop region, which is the commonest target for deletions in patients with cancers [78,79]. Somatic mutations of the mtDNA, especially in the D-loop region, were consistently found in HCC patients, indicating that qualitative alterations of mtDNA led to its decrease or increase, thus playing an important role in tumorigenic processes [21,66,67]. Wong and colleagues authored a comprehensive report focused on the relationship between mtDNA variations and the HCC clinic-pathological profile. They observed that mutations of mtDNA among 20 HCC patients increased the total amount of mtDNA compared to those without variations. Moreover, the mtDNA content was upregulated in the presence of D-loop variation and was significantly higher in three of six tumor tissues in comparison with the adjacent ones [20]. Lee et al. demonstrated that 39.3% of HCC patients harbored mutations in mtDNA, mainly due to the exacerbated oxidative damage. Additionally, 48.1% of these variations were in the mononucleotide repeat located in the polycytidine stretch between np 303–309, the (CA)n dinucleotide repeat at np 514, and a 50 bp deletion between np 298/306 and np 348/356 of the D-loop region, indicating that mutations in the D-loop structure may be involved in carcinogenesis [80]. Still, Tamori and colleagues sequenced the mtDNA of *n* = 54 HCC tissues and their non-cancerous specimens. They recognized 20 out of 52 mutation sites in the D-loop region which were specific to the HCC tissues but not to the adjacent counterparts. Moreover, a higher rate of mtDNA mutations was found in HCC compared to non-tumor tissues, and its frequency was correlated with less differentiated liver tumors, thus suggesting mtDNA mutational analysis could be a useful tool for clinical HCC prediction [81].

Controversial evidence describes the contribution of mtDNA during NAFLD progression. In a group of 45 patients with biopsy-proven NAFLD, the histological spectrum was not associated with D-loop levels, whereas hepatic methylation and transcriptional activity of the MT-ND6 were correlated with disease progression [18]. Otherwise, in 300 Turkey NASH patients, the Mt16318 C/G variant led to advanced liver injury, and the Mt16129AA and Mt16249 CC genotypes were related to fibrosis and lobular inflammation, respectively, further corroborating that D-loop nucleotide variations may be linked with NASH [12]. In another study, eighty-five variations including three deletions, six insertions, and seventy-six SNPs were detected in NAFLD patients by sequencing the entire D-loop region, reinforcing that NAFLD is a mitochondrial disease and that D-loop alterations may play a significant role into its pathogenesis [17,82].

As previously underlined, in a recent work we demonstrated that genetics significantly contributes to NAFLD pathogenesis and progression in a large cohort of NAFLD individuals. Since mitochondrial dysfunction is a key player in the switch from simple steatosis to NASH and fibrosis, we deepened the role of *PNPLA3*, *MBOAT7*, and *TM6SF2* “loss-of-function” mutations in mitochondrial dynamics by silencing *MBOAT7* (MBOAT7^−/−^) and/or *TM6SF2* (TM6SF2^−/−^, MBOAT7^−/−^TM6SF2^−/−^) in HepG2 cells carrying the *PNPLA3* variant through CRISPR/Cas9 gene editing technology [6]. The compound knock-out (KO) cells exhibited an enrichment in mitochondria with globular shape, loss of the cistern’s architecture, and ultrastructural electron density, which mirror mitochondrial degeneration. Furthermore, the double KO of *TM6SF2* and *MBOAT7* showed a shift to anaerobic glycolysis, thus supporting the synergic impact of variations on mitochondrial maladaptation during NAFLD progression [6]. By exploiting confocal microscopy, we observed that the deletion of *MBOAT7* and/or *TM6SF2* increased the total number of mitochondria, in terms of intensity of fluorescence, especially in MBOAT7^−/−^TM6SF2^−/−^ cells (Figure 1, unpublished data). Otherwise, the *spaghetti*-like/globular shape ratio of mitochondria, which reflects the physiological and damaged organelles, respectively, decreased in KO models, highlighting that the co-absence of *MBOAT7* and *TM6SF2* in the presence of *PNPLA3* augments the failed mitochondria, thus strengthening the concept that the number of mitochondria is not representative of an enhanced function (Figure 1, unpublished data). These data prompted us to investigate the genetic role in mitochondrial aberrances by assessing its possible correlation with D-loop levels. We found that the D-loop copies, which disclose the mitochondrial mass, increased in HepG2 cells silenced for *MBOAT7* and *TM6SF2* and much more in the compound knock-out model, thus emphasizing the synergic interplay between the three loss-of-function variations in mitochondrial maladaptation (Figure 1, unpublished data). Therefore, we demonstrated that genetics may contribute to the mitochondrial dysfunction which occurs during NAFLD through the modulation of mtDNA synthesis, thus proposing for the first time D-loop content as a potential biomarker in genetically predisposed NAFLD subjects.

## 6. Clinical Application of Mt-ccf: A Promising Candidate Biomarker in NAFLD Diagnosis

Notably, several fragments of mt-ccf, extracellular vesicles containing mtDNA, disrupted entire mitochondria, which originate from autophagy, apoptosis, or necrosis and could be found in body fluids [83,84]. Unfortunately, our current understanding of the biological properties of ccf-mtDNA is poor, and an accurate and efficient approach to the comprehensive profiling of mt-ccf mutations and copy numbers is still lacking. Traditional technologies, such as PCR and Sanger sequencing, face the great limitations of sensitivity and throughput in identifying heteroplasmic mtDNA mutations, and the optimization of deep-sequencing procedures enabling the accurate and efficient detection of mt-ccf are still under development [85]. Nevertheless, emerging studies indicated the diagnostic and prognostic power of mt-ccf in several disorders, such as CVD and especially in tumors [86,87]. For instance, in patients affected by both benign and malignant breast cancers, low plasma mt-ccf levels aid in discriminating the breast cancer cases from healthy individuals [86]. Ellinger et al. found that mt-ccf improved the sensitivity and specificity of ROC curves to discern prostate cancers against healthy volunteers [88], thus highlighting that assessment of mt-ccf could be attractive for identifying high-risk individuals. Finally, high mt-ccf content was associated with high death risk in 50 hospitalized HBV-HCC patients who received transarterial chemoembolization (TACE) as treatment, thus proving useful even for the outcome prediction of HCC individuals [89].

The detection of mt-ccf and other circulating biomarkers in body fluids like blood, urine, cerebrospinal fluid, or saliva is performed through liquid biopsy, which could represent an easier, cheaper, and repeatable non-invasive methodological approach for NAFLD diagnosis [90]. Karlas and colleagues demonstrated that 58 patients in whom NAFLD had been assessed through transient elastography displayed higher levels of circulating DNA compared to 13 healthy controls, and this association was strongly correlated with advanced stages of liver disease [91]. During NAFLD, the accumulated damaged mitochondria release mt-ccf, thus triggering inflammation and oxidative injury [8]. It has been observed that the plasma of NASH patients and HFD-fed mice exhibited higher levels of mt-ccf, which activated the endosomal pattern recognition receptor TLR9, required to foster inflammation and fibrosis. This leads to the upregulation of tumor necrosis factor alpha (TNF*α*) in both HFD-fed mice Kupffer cells and NAFLD patients, highlighting that serum mt-ccf exacerbates liver toxicity [14]. Consistently, Ping et al. confirmed higher levels of mt-ccf in sera of NASH patients and much more in those with advances fibrosis. Moreover, the susceptible BALB/c mice strain displayed a strong activation of hepatic stellate cells and the upregulation of pro-fibrotic factors, mainly due to an exogenous administration of mito-DAMPS, among which mt-ccf was the major component [15]. Thus, due to the drawbacks of invasive methods for NAFLD diagnosis, including sample size, repeatability, and costs which are related to its growing incidence, the detection of mt-ccf through liquid biopsy may be a valid alternative to assess NAFLD.

As shown in the previous paragraph, our preliminary data demonstrated that D-loop copies were increased in HepG2 cells silenced for both *MBOAT7* and *TM6SF2* genes, paralleling mitochondrial dysfunction in term of mass, morphology, and function (Figure 1). We then investigated whether the *PNPLA3*, *MBOAT7,* and *TM6SF2* variations may trigger the release of mt-ccf in our genetic in vitro models by extracting short DNA fragments (<100 bp) from cell supernatant. Mt-ccf was measured using quantitative real time PCR (qRT-PCR) by the amplification of the cytochrome c oxidase subunit III (COXIII) and Mitochondrially Encoded NADH: Ubiquinone Oxidoreductase Core Subunit 1 (ND1) genes. Interestingly, we found that HepG2 cells silenced for *MBOAT7* and *TM6SF2* exhibited increased levels of COXIII and NDI content, which was exacerbated in the compound knock-out (Figure 1). To summarize, it is plausible that a high number of D-loop copies is not representative of enhanced mitochondrial function, although they parallel the accumulation of failed mitochondria, which in turn release mt-ccf.

## 7. Concluding Remarks

Over the last two decades, NAFLD has become the commonest chronic liver disorders affecting both adults and children worldwide. Until now, no therapeutic approaches have been approved for NAFLD treatment, and lifestyle interventions remain the only strategy for patients’ care. Currently, hepatic biopsy is the gold standard for NAFLD diagnosis, although its invasiveness, high cost, and poor repeatability pave the way for the identification of novel non-invasive biomarkers. NAFLD is a multifactorial disease in which both environmental and genetic factors interact in its onset and progression, and it encompasses a wide *spectrum* of metabolic disorders ranging from simple steatosis up to NASH, fibrosis, and HCC. Thus, several studies focused on the development of an NAFLD risk prediction model based on biochemical indicators including high-density lipoprotein cholesterol (HDL-c), total cholesterol, alanine aminotransferase (ALT), metabolic disorders like obesity and T2DM, and lifestyle features such as dietary habits and physical activity. Additionally, it has now well-established that genetics strongly impact the overall NAFLD *spectrum*, and nowadays, polymorphisms in the *PNPLA3*, *MBOAT7*, and *TM6SF2* genes act as the main NAFLD genetic predictors. Several studies have recommended the utility of polygenic risk score to foresee NAFLD development and severity, and their combination with dynamic risk markers could improve the prognostic accuracy.

To date, mitochondria are considered one of the key players in NAFLD pathogenesis and represent a promising new avenue for NAFLD diagnosis. During NAFLD, in response to higher fat accumulation, the liver runs into a mechanism called “mitochondria flexibility” through which these organelles modify number, biomass, and activity by modulating fusion, fission, and mitophagy. Nonetheless, this adaptability is lost during NAFLD progression, resulting in the imbalance of the mitochondrial lifecycle, low levels of mitophagy, and accumulation of failed mitochondria with irregular shape and compromised OXPHOS capacity. Consequently, the exacerbated ROS production triggers the inflammatory and fibrotic response and in turn damages the mtDNA. Emerging evidence suggests that variations in the non-coding D-loop region impair the mtDNA replication and transcription, resulting in a higher number of mitochondria, which does not necessarily entail increased function. The high quantity of D-loop copies is correlated with the severity of liver disease, thus underlining its prognostic value. Additionally, the loss of mitochondrial dynamics together with low levels of mitophagy sustain the accumulation of damaged mitochondria which, in turn, prompt the release of mt-ccf, exacerbating liver injury (Figure 2). The mt-ccf content could be detected in body fluids through liquid biopsy, which represents a novel, non-invasive, cheaper, and repeatable NAFLD diagnostic approach.

The important contribution of both genetics and mitochondrial dysfunction in NAFLD is already well-known. Nevertheless, the impact of the main genetic predictors on these organelles has been poorly investigated. Our recently published data highlighted that the co-presence of *PNPLA3*, *MBOAT7*, and *TM6SF2* loss-of-functions in hepatocytes impairs mitochondrial adaptability by increasing the total number of mitochondria, among which the majority display a globular non-physiological shape. The D-loop copies, which disclose the number of mitochondria, are consistently augmented in knock-out cells alongside the release of mt-ccf. Therefore, it could be speculated that a risk score including genetics as well as biochemical parameters, metabolic comorbidities, D-loop variations, and mt-ccf levels might improve diagnostic accuracy by exploiting ROC curve analysis to estimate its predictivity and sensitivity. To conclude, our findings, together with the emerging literature, suggest that variations in the D-loop region and impaired content of mt-ccf may predict the onset and progression of NAFLD, thus opening a spyhole into their use as non-invasive biomarkers.

## Figures and Tables

**Figure 1 ijms-24-02331-f001:**
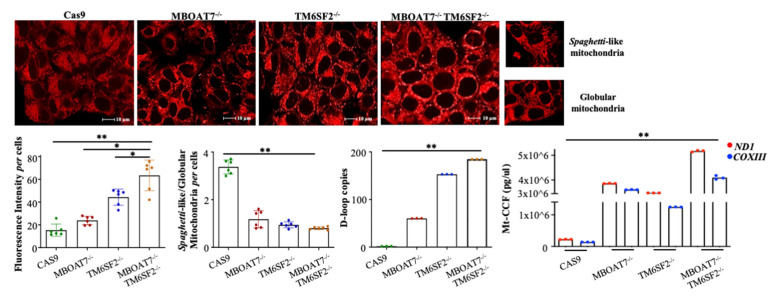
The *PNPLA3*, *MBOAT7*, and *TM6SF2* loss-of-functions impair the mitochondrial number and morphology in genetic NAFLD in vitro models: Mitochondria were stained in untreated Cas9, MBOAT7^−/−^, TM6SF2^−/−^, and MBOAT7^−/−^TM6SF2^−/−^ cells by using Mitotracker red. Florescence intensity per cell and *spaghetti*-like/globular shape ratio per cell were quantified by ImageJ software in 10 random non-overlapping micrographs. The number of mitochondria in terms of intensity of fluorescence increased in MBOAT7^−/−^, TM6SF2^−/−^ cells and much more in MBOAT7^−/−^TM6SF2^−/−^ cells, while the *spaghetti*-like/globular shape ratio decreased in knock-out models (** *p* < 0.01 vs. Cas9, * *p* < 0.05 vs. MBOAT7^−/−^ and TM6SF2^−/−^). D-loop copies were quantified in total DNA extracted from untreated Cas9, MBOAT7^−/−^, TM6SF2^−/−^, and MBOAT7^−/−^TM6SF2^−/−^ cells and were strongly increased in TM6SF2^−/−^ and MBOAT7^−/−^TM6SF2^−/−^ models (** *p* < 0.01 vs. Cas9). Amount (pg/μL) of ND1 (red dots) and COXIII (blue dots) was quantified in untreated Cas9, MBOAT7^−/−^, TM6SF2^−/−^, and MBOAT7^−/−^TM6SF2^−/−^ cell supernatants and was significantly upregulated in all knock-out models (** *p* < 0.01 vs. Cas9).

**Figure 2 ijms-24-02331-f002:**
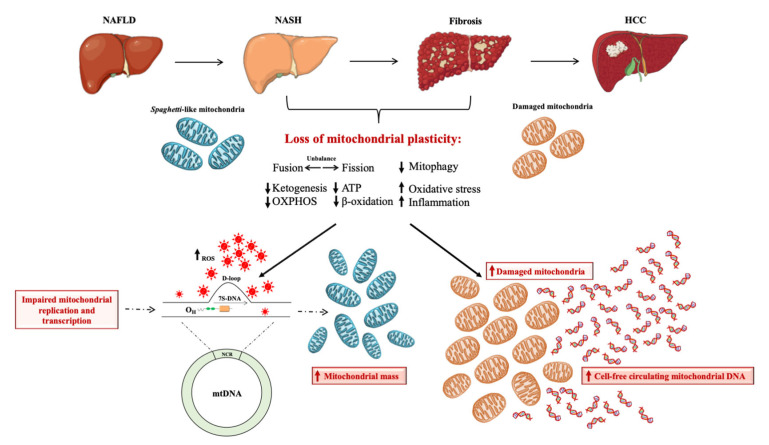
D-loop and mt-ccf: markers of mitochondrial dysfunction during NAFLD progression. In the early stage of NAFLD, in response to higher fat accumulation, the liver engages in a strategy called “mitochondrial flexibility” through which mitochondria adapt their number, mass, and activity by modulating the lifecycle and turnover. Nonetheless, this adaptability is lost during NASH progression, resulting in the leak of mitochondrial plasticity and leading to the imbalance of fusion and fission processes, which parallels the reduction in mitophagy. This results in the failure of mitochondrial activity in terms of lower ketogenesis, OXPHOS capacity, ATP production, and β-oxidation along with the augmented oxidative stress and inflammation. The exacerbated ROS production fosters mtDNA damage in the D-loop region, thus impairing its replication and transcription and boosting the mitochondrial mass. Additionally, the fusion–fission imbalance parallels lower mitophagy, heightening the release of mt-ccf from damaged mitochondria, which in turn triggers the inflammatory and oxidative response. Thus, the increased mitochondrial biomass or number along with the higher levels of mt-ccf are not representative of enhanced mitochondrial function and could be considered as a potential biomarker of NAFLD progression.

**Table 1 ijms-24-02331-t001:** Associations between mtDNA mutations and NAFLD *spectrum*.

mt-DNA Region	Mutations	NAFLD Stages	Number of Cases	Other Diseases	References
**-loop**	94 variations: 2 deletions, 4 insertions and 88 single nucleotide polymorphisms	Steatosis, NASH, fibrosis (NAFLD)	43 NAFLD patients	-	Kamfar S. et al., 2019 [17]
**D-loop**	m.16318 A>C, CC	NASH	150 NASH patients	MetS, T2DM, and hypothyroidism	Hasturk B et al., 2019 [12]
**D-loop**	cytosine methylation (5mC)	NASH	45 NAFLD patients	-	Pirola CJ et al., 2013 [18]
**D-loop**	m.16129 G>A, AA	Fibrosis Cirrhosis	150 NASH patients	-	Hasturk B et al., 2019 [12]
**D-loop**	m.16249 T>C, CC	Steatosis Lobular inflammation	150 NASH patients	-	Hasturk B et al., 2019 [12]
**D-loop**	polycytidine stretch between np 303–309	HCC	61 HCC patients: 40.7%	-	Lee HC et al., 2004 [19]
**D-loop**	(CA)n dinucleotide repeat at np 514	HCC	61 HCC patients: 37.0% (10/27)	-	Lee HC et al., 2004 [19]
**D-loop**	50bp deletion between np 298/306	HCC	61 HCC patients: 78%	-	Lee HC et al., 2004 [19]
**D-loop**	23 somatic mutations	HCC	20 HCC patients: 50%	Other tumors (breast, colon, thyroid cancers)	Wong LJ et al., 2004 [20]
**D-loop**	21 substituitions CCCC insertion at 573 position	HCC	86 HCC patients: 30.43% (14/46)	-	Qiao L et al., 2017 [21]
**NADH dehydrogenase**	MT-ND6 (cytosine methylation (5mC))	NASH	45 NAFLD patients	-	Pirola CJ et al., 2013 [18]
**NADH dehydrogenase**	MT-ND4 (m.11040 T > C) MT-ND2 (m.4769 A > G)	NASH	64 NAFLD patients	-	Sookoian S et al., 2016 [22]
**MT-ATP6, MT-CYB, MT-CO1**	MT-ATP6 (m.9101 T > C) MT-CYB (m.14766 C > T) MT-CO1 (m.7028 C > T)	NASH	64 NAFLD patients	-	Sookoian S et al., 2016 [22]
**NADH dehydrogenase**	MT-ND5 (m.13676 A>G)	HCC	86 HCC patients: 19.57% (9/46)	-	Qiao L.+ et al., 2017 [21]
**NADH dehydrogenase**	Heteroplasmic mutations (MT-ND1, MT-ND3, MT-ND4, MT-ND5, MT-ND6)	HCC	88 HCC patients	-	Li W et al.,2017 [23]

## Data Availability

Not applicable.

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
