# Peer review of "The Non-Invasive Assessment of Circulating D-Loop and mt-ccf Levels Opens an Intriguing Spyhole into Novel Approaches for the Tricky Diagnosis of NASH"

_ijms, 2023, doi:10.3390/ijms24032331_

Round 1
Reviewer 1 Report
The manuscript by Paolini E. et al, is well reviewed and provided meaningful information on mitochondrial dynamics and non-invasive biomarkers in predicting NAFLD onset and progression, based on the authors' findings and a multitude of other information including the latest literature.
No major comments, but minor comments are
1) I would like to see a comprehensive table showing research results on NAFLD onset/progression and mtDNA-containing biomarkers.
2) Although there are many significant references provided, many of them (e.g., references 3, 4, 8, 9, 10, 13, 16, etc.) do not list the name of the journal from which the references are derived, so please check.
Author Response
Reviewer 1
We would thank the Reviewer for the positive comments.
The manuscript by Paolini E. et al, is well reviewed and provided meaningful information on mitochondrial dynamics and non-invasive biomarkers in predicting NAFLD onset and progression, based on the authors' findings and a multitude of other information including the latest literature.
No major comments, but minor comments are
1) I would like to see a comprehensive table showing research results on NAFLD onset/progression and mtDNA-containing biomarkers.
We thank the Reviewer for the suggestion. We have now improved the manuscript by introducing Table 1 which summarizes the current associations between mtDNA mutations and NAFLD stages.
2) Although there are many significant references provided, many of them (e.g., references 3, 4, 8, 9, 10, 13, 16, etc.) do not list the name of the journal from which the references are derived, so please check.
We have checked all the references as required by the Reviewer.
Reviewer 2 Report
Review of manuscript entilted: “The non-invasive assessment of circulating D-loop and mt-ccf levels opens an intriguing spyhole into novel approaches for the tricky diagnosis of NASH” authored by Erika Paolini, Miriam Longo, Alberto Corsini and Paola Dongiovanni.
First of all thank you for possibility to review this interesting manuscript.
In the presented review authors present current state of knowledge about nonalcoholic liver diseases (NALFD, NASH and HCC) with particular emphasis on D-loop and mt-ccf.
In the introduction authors present some statistics about prevalence and basic information about undertaken problem. Introduction is ended with well-defined aim of this review. Then, authors procced with reviewing more advanced topic with very detailed manner.
In my opinion, manuscript is well-written and moderately easy to follow, it covers its topic in detail with some additional results obtained by authors, which were previously unpublished. Cited literature seems to be very recent except some publications. Both figures are really informative and easy to read, however I have one remark about Figure 1.
Major concerns:
None
Minor concerns:
- Please try to avoid using abbreviations in the title
- In my opinion authors should consider creating separate chapter about genetic markers and risk factors (genotypes), which is now included in chapter 2. “NAFLD diagnosis and prognosis: currently drawbacks and new-fashioned strategies”
- Line 283 – missing space after dot
- Figure 1 BCDE - x-axis and y-axis labels could be larger since they are hard to decipher
Author Response
Reviewer 2
Review of manuscript entilted: “The non-invasive assessment of circulating D-loop and mt-ccf levels opens an intriguing spyhole into novel approaches for the tricky diagnosis of NASH” authored by Erika Paolini, Miriam Longo, Alberto Corsini and Paola Dongiovanni.
First of all thank you for possibility to review this interesting manuscript.
In the presented review authors present current state of knowledge about nonalcoholic liver diseases (NALFD, NASH and HCC) with particular emphasis on D-loop and mt-ccf.
In the introduction authors present some statistics about prevalence and basic information about undertaken problem. Introduction is ended with well-defined aim of this review. Then, authors procced with reviewing more advanced topic with very detailed manner.
In my opinion, manuscript is well-written and moderately easy to follow, it covers its topic in detail with some additional results obtained by authors, which were previously unpublished. Cited literature seems to be very recent except some publications. Both figures are really informative and easy to read, however I have one remark about Figure 1.
Major concerns:
None
Minor concerns:
We thank the Reviewer for the enthusiastic comments about the manuscript and we strongly appreciate his/her suggestions to improve it.
- Please try to avoid using abbreviations in the title
We agree with the Reviewer and we have deleted the abbreviation in the title.
- In my opinion authors should consider creating separatechapter about genetic markers and risk factors (genotypes), which is now included in chapter 2. “NAFLD diagnosis and prognosis: currently drawbacks and new-fashioned strategies”
We thank the Reviewer for the suggestion. We have slimmed down paragraph 2 “NAFLD diagnosis and prognosis: currently drawbacks and new-fashioned strategies” by inserting the sub-chapter 2.1 entitled “Genetics and metabolic factors co-aid NAFLD diagnosis”.
- Line 283 – missing space after dot
We have corrected all the typos.
- Figure 1 BCDE - x-axis and y-axis labels could be larger since they are hard to decipher
We have now enlarged the x- and y-axis labels.